# Efficacy of Phytochemicals Derived from Roots of *Rondeletia odorata* as Antioxidant, Antiulcer, Diuretic, Skin Brightening and Hemolytic Agents—A Comprehensive Biochemical and In Silico Study

**DOI:** 10.3390/molecules27134204

**Published:** 2022-06-30

**Authors:** Anjum Khursheed, Saeed Ahmad, Kashif-ur-Rehman Khan, Muhammad Imran Tousif, Hanan Y. Aati, Chitchamai Ovatlarnporn, Huma Rao, Umair Khurshid, Bilal Ahmad Ghalloo, Sobia Tabassum, Abdul Basit

**Affiliations:** 1Department of Pharmaceutical Chemistry, Faculty of Pharmacy, The Islamia University of Bahawalpur, Bahawalpur 63100, Pakistan; anjumkhursheedrana@gmail.com (A.K.); rsahmed_iub@yahoo.com (S.A.); humarao80@gmail.com (H.R.); umairch.k@gmail.com (U.K.); drbilal29@hotmail.com (B.A.G.); sobiat350@gmail.com (S.T.); 2Department of Chemistry, Division of Science and Technology, University of Education Lahore, Lahore 54770, Pakistan; imran.tousif@ue.edu.pk; 3Department of Pharmacognosy, College of Pharmacy, King Saud University, Riyadh 11495, Saudi Arabia; 4Department of Pharmaceutical Chemistry, Faculty of Pharmaceutical Sciences, Prince of Songkla University, Hat Yai 90110, Thailand; chitchamai.o@psu.ac.th; 5Quaid-e-Azam College of Pharmacy, Quaid-e-Azam Educational Complex, Sahiwal 54000, Pakistan; abdulbasit.pharmd@gmail.com

**Keywords:** *Rondeletia odorata*, polyphenols, flavonoids, antioxidants, enzyme inhibition studies, hemolytic activity, LC-MS, docking studies

## Abstract

Roots of *Rondeletia odorata* are a rich source of phytochemicals with high antioxidant potential and thus may possess health benefits. This study used the LC-MS technique to identify phytoconstituents in *R. odorata* roots extract/fractions. Results revealed that *n*-butanol fraction and ethanolic extract contained total phenolic and flavonoid contents with values of 155.64 ± 0.66 mgGAE/g DE and 194.94 ± 0.98 mgQE/g DE, respectively. Significant potential of antioxidants was observed by DPPH, CUPRAC and FRAP methods while the ABTS method showed moderate antioxidant potential. Maximum % inhibition for urease, tyrosinase and carbonic anhydrase was shown by ethanolic extract (73.39 ± 1.11%), *n*-butanol soluble fraction (80.26 ± 1.59%) and ethyl acetate soluble fraction (76.50 ± 0.67%) which were comparable with thiourea (standard) (98.07 ± 0.74%), kojic acid (standard) (98.59 ± 0.92%) and acetazolamide (standard) (95.51 ± 1.29%), respectively, while all other extract/fractions showed moderate inhibition activity against these three enzymes. Hemolytic activity was also observed to range from 18.80 ± 0.42 to 3.48 ± 0.69% using the standard (triton X-100) method. In total, 28 and 20 compounds were identified tentatively by LC-MS analysis of ethanolic extract and *n*-butanol soluble fraction, respectively. Furthermore, molecular docking was undertaken for major compounds identified by LC-MS for determining binding affinity between enzymes (urease, tyrosinase and carbonic anhydrase) and ligands. It was concluded that active phytochemicals were present in roots of *R. odorata* with potential for multiple pharmacological applications and as a latent source of pharmaceutically important compounds. This should be further explored to isolate important constituents that could be used in treating different diseases.

## 1. Introduction

Over the past few years, medicinal plants have been explored extensively due to the presence of a vast variety of secondary metabolites to discover the lead compounds which can contribute to different pharmacological and therapeutic efficacies [1]. Approximately 25% of various therapeutical moieties used at present times have been obtained from plants [2]. The World Health Organization (WHO) validates that more than 80% of world’s total population depends on different plant products to meet their basic healthcare needs [3]. Different therapeutic activities, for example, antioxidant, anticancer, insecticidal, antibacterial, antiviral, antiaging, antifungal, antimalarial and anti-inflammatory, etc., of plants depend upon the presence of a vast variety of secondary metabolites that are separated by different advanced, sensitive, and sophisticated techniques [4]. For this purpose, around 20,000 various plant species had been investigated for therapeutical purposes [5].

Environmental stress conditions, smoke, chemicals and drugs or aerobic cellular metabolism are among those exogenous factors that contribute to the formation of Reactive Oxygen Species (ROS) [6]. The accumulation of these reactive species in the body results in initiation of drastic chain reactions that ultimately destroy many vital biological components that are carbohydrates, lipids, proteins, and DNA [7]. Thus, these species are associated with causing different diseases, e.g., cardiovascular diseases, atherosclerosis, Alzheimer’s disease, Huntington’s disorder, Parkinson’s disease, insulin resistance, diabetes mellitus and some kinds of cancer [8]. Therefore, they represent promising targets for the drug treatment of various pathological conditions. In this domain, natural antioxidants are getting increasing attention. They are serving as novel lead compounds for manufacturing of new drugs and are also representing an alternative to the use of synthetic antioxidants such as butylhydroxytoluene (BHT), butylhydroxyanisole (BHA), or propylgallate in food technology [9]. According to scientific research papers, two-third of all plants have been reported for their antioxidant potential and medicinal value [10].

Urease is a nickel-containing metallo-enzyme [11] which neutralizes stomach acid and abnormally elevates pH at a higher level, resulting in the survival of pathogenic bacterium *H. pylori*. It may cause gastrointestinal diseases, peptic and duodenal ulcers, and gastric cancer [12]; while the urease presence itself may lead to urinary stones [13]. Therefore, ureases have become important targets for research both in human and animal health, as well as in agriculture [14]. Hence, urease inhibitors discovery has the utmost importance [15] and many inhibitors have been described in the past but were prevented in vivo because of their toxicity or instability. Therefore, there are unmet medical needs for novel and efficacious urease inhibitors with greater stability and low toxicity [16].

Browning and hyperpigmentation are two common undesirable phenomena for human skin and tyrosinase has been recognized as responsible for these two phenomena in mammals [17]. This led the scientists to identify, isolate, synthesize and characterize new potent tyrosinase inhibitors [4]. Very few inhibitors are in use for clinical purposes and as skin-whitening agents. As the demand for tyrosinase inhibitors increases both in clinical and industrial fields, improved screening techniques are also undergoing rapid development for tyrosinase inhibitors and putative skin-whitening agents [18].

Carbonic anhydrases (CAs) play an important role in equilibrating the chemical reaction among bicarbonate, carbon dioxide and protons. These simple molecules/ions are essential for many physiological processes throughout the tree of life [19]. CAs inhibition serves many pharmacologic functions, for example, it can be used as diuretics, or can treat and prevent various diseases such as glaucoma, mountain sickness, epilepsy, CHF, peptic ulcers, neurological disorders and osteoporosis, as well as can be used as diagnostic tools [20]. Many synthetic CA inhibitors have been prepared and evaluated over the last few decades, whereas naturally occurring CAI compounds are going to be investigated soon [21].

Rubiaceae is one of the largest families of angiosperms, comprises 660 genera and 13,200 species and is found all over the world [22]. Many of the plants have widespread use in folk medicine and some showed anti-inflammatory, analgesic, antibacterial, mutagenic, antiviral and antioxidant effects on vascular diseases as well as had activity on the central nervous system [23]. *Rondeletia odorata* Jacq. (Syn: *R. speciosa* Lodd; *R. brilliantissima* Hend; *R. coccinea* and *R. obovata* L.) [24] belongs to the family Rubiaceae, is an evergreen shrub native to Cuba and Panama but is also grown in gardens in Pakistan. Common names include “Sweet Smelling rondeletia” and “fragrant Panama rose”. Various plants of the *genus Rondeletia* have been used traditionally in different countries around the world [25]. The present work is the first step aiming to observe the preliminary phytochemical screening, pharmacological assays in vitro and molecular docking of *R. odorata* Jacq. roots extract/fractions as an alternate source of antiulcer, diuretic, skin brightening and antioxidant agents. This systematic study represents the first step towards evaluating the pharmacological potential of this plant so that it can be brought into the commercial health market to serve the community with its potential benefits.

## 2. Results

### 2.1. Phytochemical Analysis

#### 2.1.1. Preliminary Phytochemical Profiling

Preliminary phytochemical studies of ethanolic extract (ROEE), *n*-hexane soluble fraction (ROHF), ethyl acetate soluble fraction (ROEF), *n*-butanol soluble fraction (ROBF) and water soluble fraction (ROWF) of roots of *R. odorata* were performed. These studies showed the presence of primary and secondary metabolites (Table 1). Among primary metabolites, carbohydrates and amino acids were observed to be present in ROBF, while proteins were identified in ROHF. Lipids were found to be in abundant amounts and were present in ROEE, ROEF and ROBF. Among secondary metabolites, alkaloids and flavonoids were identified as abundant in all extracts/fractions. Phenols, tannins and saponins were observed in moderate amounts whereas glycosides were not found in any extract/fractions.

#### 2.1.2. Total Phenolic Contents (TPC)

The highest amount of TPC was observed in ROBF with 155.64 ± 0.66 mg gallic acid equivalent/g of dry extract while the lowest amount was observed in ROWF with 30.70 ± 0.99 mg gallic acid equivalent/g of dry extract. TPC contents values showed the pharmacological importance of the plant (Figure 1).

#### 2.1.3. Total Flavonoid Contents (TFC)

The highest amount of TFC was found in ROHF with 256.10 ± 1.02 mg quercetin equivalent/g of dry extract and the lowest amount was observed in ROWF with 55.77 ± 0.81 mg quercetin equivalent/g of dry extract. TFC content values showed the biological potential of the plant (Figure 1).

### 2.2. In Vitro Pharmacological Potential

The pharmacological potential (in vitro) of extract/fractions of roots of *R. odorata* was determined by performing antioxidant assays, enzyme inhibition activities and hemolytic activity.

#### 2.2.1. Antioxidant Analysis

Radical Scavenging Antioxidant Assay

Radical scavenging antioxidant potential evaluated by ABTS and DPPH was ordered as follows: ROEF > ROBF > ROEE > ROWF > ROHF for ABTS and ROEE > ROBF > ROEF > ROHF > ROWF for DPPH. The maximum radical scavenging potential determined for ABTS was of ROEF with 87.92 ± 1.44 mg trolox equivalent/g of dry extract while the minimum potential was of ROHF with 49.25 ± 1.42 mg trolox equivalent/g of dry extract. The maximum antioxidant potential for DPPH was of ROEE with 197.85 ± 1.42 mg trolox equivalent/g of dry extract while the minimum potential was of ROWF with 51.47 ± 0.72 mg trolox equivalent/g of dry extract (Table 2).

Reducing Power Antioxidant Assays

These assays were determined by FRAP and CUPRAC and were ordered as follows: ROBF > ROEE > ROEF > ROHF > ROWF for FRAP and ROHF > ROBF > ROEE > ROEF > ROWF for CUPRAC. The maximum reducing potential determined for FRAP was of ROBF with 239.92 ± 1.72 mg trolox equivalent/g of dry extract whereas the minimum potential for FRAP was of ROWF with 150.07 ± 1.59 mg trolox equivalent/g of dry extract. The maximum reducing potential determined for CUPRAC was of ROHF with 312.77 ± 1.03 mg trolox equivalent/g of dry extract while the minimum potential was of ROWF with 145.26 ± 0.57 mg trolox equivalent/g of dry extract (Table 2).

#### 2.2.2. Enzyme Inhibition Assays

Urease Inhibition Potential

Urease inhibition potential of different extract/fractions of roots of *R. odorata* was determined by a method mentioned in [26] with slight modifications. Urea was taken as the substrate and results were elaborated as % inhibition ± standard deviation and ordered as follows: ROEE > ROBF > ROEF > ROHF > ROWF. Maximum % inhibition was observed for ROEE (73.39 ± 1.11%) and minimum % inhibition was observed by ROWF (45.69 ± 0.71%). Urease inhibition results for different extract/fractions showed roots of *R. odorata* as a potential inhibitor of urease enzyme (Table 3).

Tyrosinase Inhibition Potential

Tyrosinase inhibition potential for roots of *R. odorata* was determined by [27] with minor modifications. The results were expressed as % inhibition ± standard deviation. The % inhibition of tyrosinase enzyme was ordered as follows: ROBF > ROEE > ROEF > ROHF > ROWF. The maximum inhibition was observed for ROBF (80.26 ± 1.59%) which was comparable with % inhibition of kojic acid (standard) (98.59 ± 0.92%). The % inhibition for all extract/fractions of roots of *R. odorata* was in the range of 80.26–43.33%, which showed these plant roots as a potent tyrosinase enzyme inhibitor (Table 3).

Carbonic Anhydrase (CA) Inhibition Potential

CA enzyme inhibition potential was determined by [28] with some modifications. 4-Nitrophenol acetate served as substrate and acetazolamide was the standard. The % inhibition potential of different extract/fractions was in the order: ROEF > ROEE > ROBF > ROHF > ROWF. The maximum % inhibition was shown by ROEF (76.50 ± 0.67%) which was nearly equal to acetazolamide (standard) (95.51 ± 1.29%) whereas minimum % inhibition was shown by ROWF (51.60 ± 1.13%). The % inhibition range for all the extract/fractions was between 76.50 and 51.60% which showed these plant roots as a potential diuretic (Table 3).

#### 2.2.3. Hemolytic Potential

Data shown in (Table 4) represented the hemolytic potential of different extract/fractions of roots of *R. odorata*. The hemolytic % was in the order: ROEE > ROHF > ROWF > ROEF > ROBF. The value of maximum hemolytic % was 18.80 ± 0.42% for ROEE while the value of minimum hemolytic % was 3.48 ± 0.69% for ROBF. All five extracts/fractions showed hemolysis activity less than 30%, so all fractions are nontoxic and safe as food.

### 2.3. UHPLC-ESI-QTOF-MS Analysis

The analyses of polar regime, i.e., ROEE and ROBF were carried out in positive ionization mode which resulted in the identification of the presence of phenolics, flavonoids and other secondary phytoconstituents. In these analyses, complex chromatograms (Figure 2 and Figure 3) were obtained with a matching score >98. In total, 28 and 20 compounds were identified in ROEE and ROBF, respectively (Table 5 and Table 6).

In ROEE, among phenolic compounds, artemidinol possesses antithrombolytic and anticarcinogenic activities [29]. Xanthone has antimicrobial, antioxidant and cardio-protective effects [30], while 2-O-Feruloylhydroxycitric acid exhibits strong antioxidant activity [31]. Hydrojuglone glucoside has antitrypanosomal activity [32] and Ligustroside has antiviral and anti-inflammatory potential [33]. 3′-Glucosyl-2′,4′,6′-trihydroxyacetophenone [34] and Aloesol 7-glucoside [35,36] have antioxidant and antibacterial activities. Among flavonoid compounds, Glyflavanone A has antioxidant and antiulcer activities [37]. Ponganone III is a chemopreventive agent [38] whereas 3,7,8,4′-tetrahydroxyflavone possesses antiparasitic activity [39]. 2-O-Caffeoylglucarate is a strong antioxidant agent [40]. 1-Caffeoyl-4-deoxyquinic acid has antiacetylcholinesterase and antibutyrylcholinesterase activities [41]. Hosloppin has antidiabetic potential [42]. Tetramethylquercetin 3-rutinoside has antioxidant and anticancer potential [43]. Other secondary metabolites, which belong to different classes, occupy antioxidant, antibacterial, antifungal, antiplasmodial, and anti-HIV activities.

In ROBF, phenolics, flavonoids and alkaloids along with other secondary metabolites were identified. Among phenolic contents, Avenanthramide 1c has antioxidant and anti-inflammatory potential [44]. 5-O-Methylleridol showed antimicrobial and anticancer activities [45] while Lophirone E has gametocytocidal antimalarial potential [46]. Among flavonoid constituents, 4′,5,6,7,8-Pentahydroxy-3′-methoxyflavone has antiinflammatory, antioxidant potential and treats cardiovascular diseases [47] while isosinensetin shows antioxidant and antihemolytic activities [48]. Among alkaloids, Robustine exhibits antileishmanial and antitrypanosomal activities [49]. Flazin is an antidiabetic agent [50]. Pteleine shows antitumor and antimicrobial activities [51]. Oxonantenine possesses cytotoxic activity [52] while gamma-Fagarine has antitrichomonas potential [53]. Erysopine has antifeedant potential [54].

UHPLC-ESI-QTOF-MS analysis of polar extract/fraction (ROEE and ROBF) also confirmed the occurrence of various quinones, diterpene lactones and ketones (Table 5 and Table 6). The presence of these very important bioactive metabolites suggests the use of roots of *R. odorata* in nutraceuticals and food supplements.

### 2.4. In Silico Molecular Docking Studies

To look better into the inhibition potential of understudy compounds and to compare this data with enzyme inhibition findings, 14 compounds from the liquid chromatography–mass spectrometry (LC-MS) profile of ROEE and ROBF were docked against urease, tyrosinase and carbonic anhydrase proteins. The maximum binding affinity was shown by Glyflavanone A, i.e., −9 in the case of urease enzyme while binding affinity shown by thiourea (standard) was −3.4 (Table 7, Figure 4 and Appendix A). The maximum binding affinity in the case of tyrosinase was shown by hosloppin, i.e., −10 while it was −5.9 shown by standard kojic acid (Table 8, Figure 5 and Appendix A). In the case of carbonic anhydrase, hosloppin showed the highest binding affinity, i.e., −7.8 while the binding affinity exhibited by acetazolamide (standard) was −6.4 (Table 9, Figure 6 and Appendix A).

## 3. Discussion

Phytochemical analysis is very important for evaluating the possible medicinal utilities of a plant and also to determine the active principles responsible for the known biological activities exhibited by the plants. Further, it provides the base for targeted isolation of compounds and to perform more precise investigations [55]. The phytochemical screening of the extract/fractions of roots of *R. odorata* demonstrated that extract/fractions are the ultimate source of tannins, saponins, flavonoids, lipids, alkaloids and phenols. Secondary metabolites, for example, alkaloids, possess antimicrobial and analgesic activities; tannins and flavonoids demonstrate as antibacterial and antioxidant agents [56], while saponins act as anti-diabetic, anticancer, antibacterial and anti-inflammatory agents [57]. These phytoconstituents’ presence in the extract/fractions of roots of *R. odorata* might be a reason of its therapeutic efficacy.

The highest total phenolic contents of *n*-butanol fraction, calculated from the calibration curve (*R*^2^ = 0.999), was 155.64 ± 0.66 gallic acid equivalent/g of dry extract, and the highest total flavonoid content of *n*-hexane fraction (*R*^2^ = 0.998) was 256.10 ± 1.02 quercetin equivalent/g of dry extract (Figure 1). Redox properties have been exhibited by phenolic compounds and are responsible for their antioxidant potential [58]. These are the hydroxyl groups which impart radical scavenging potential to phenolic compounds so the total phenolic contents might be considered as a basis for antioxidant activity. Flavonoids, such as flavanols, flavones and condensed tannins are phytoconstituents of prime importance and their antioxidant potential depends on the presence of free hydroxyl groups, especially 3-OH. Flavonoids contain antioxidant activity and can be used in both in vitro and in vivo studies [59,60]. As this is the first report on the phenolic and flavonoid profile of roots of *R. odorata*, thorough isolation and identification of constituents should be undertaken to identify the active phenolic and flavonoid components.

The common products of metabolic processes are reactive oxygen species (ROS). Excessive ROS accumulation has various adverse effects on lipids, proteins and DNA, which resulted in inflammation and tissue injury [61]. To enhance efficiency of the immune system, different antioxidants should be used to detoxify these reactive species. Synthetic origin antioxidants are given less importance as compared to natural antioxidants because of their adverse effects. Medicinal plants which are used globally for their therapeutic potential are a bigger source of natural-origin antioxidant agents [62]. Polyphenols are important and biologically active components of plants and their consumption resulted in producing various therapeutic effects, such as anticancer, antidiabetic, antibacterial, antiviral and antioxidant [63]. Anticancer, antiallergic, anti-inflammatory and antioxidant are among the biological effects which are shown by flavonoid compounds [64]. As the previous research studies showed, there is a direct connection between phenolic compounds and antioxidant activity [65]. To the best of our knowledge, there is no report available on the antioxidant activity of the ethanolic extract, *n*-hexane soluble fraction, ethyl acetate soluble fraction, *n*-butanol soluble fraction and water soluble fraction of roots of *R. odorata*. Extracts/fractions with greater flavonoid and phenolic contents exhibited significant antioxidant activities (Table 2) [66].

Urease is product of *Helicobacter pylori*, which is a causative agent of gastroduodenal diseases resulting in peptic and gastric cancer. Urease minimizes the stomach acidity by converting urea into ammonia in the stomach. This low acidic media provides an ideal growth condition to *H. pylori* and enhances its colonization. Urease is also a virulence factor in urinary tract infections and gastrointestinal infections in animals and humans. *H. pylori* is sensitive towards antibiotics, but treatment failure occurs in more than 15% of patients. The alternate choice of urease inhibition to treat *H. pylori* infection is natural products [67]. The search for urease inhibitors, with better therapeutic efficacy, bioavailability and lesser side effects, is ongoing. Our research regarding the urease inhibition potential of roots of *R. odorata* revealed an extremely potent inhibitor of this enzyme. Ethanolic extract (ROEE) and *n*-butanol soluble fraction (ROBF) of roots showed significant inhibition (73.39 ± 1.11% and 70.29 ± 0.81% inhibition, respectively) while moderate to minimum results were shown using the ethyl acetate soluble (ROEF), *n*-hexane soluble (ROHF) and water soluble fractions (ROWF) (66.36 ± 0.91%, 53.97 ± 1.63% and 45.69 ± 0.71% inhibition, respectively). Such significant results of urease inhibition may be due to the presence of bioactive constituents as demonstrated by LC-MS profile, such as Glyflavanone A (Table 7), which showed maximum binding interaction with urease enzyme, and may be due to some other compounds in these extracts/fractions.

Tyrosinase has an important role in melanin production. Melanin overproduction results in melasma and age spots. Tyrosinase inhibitors and antioxidants agents are desired skin-protecting agents in the food and cosmetics industry [68]. Over time, many skin-whitening products have been introduced into the market but none have been found to be satisfactory due to their toxicity and mutagenic effects as observed for hydroquinone [69]. Newer tyrosinase inhibitors from natural origin with better therapeutic efficacy, skin penetration and lesser side effects are still being identified. Our research regarding the tyrosinase inhibition potential of roots of *R. odorata* revealed an extremely potent inhibitor of this enzyme. Significant inhibition results were shown by *n*-butanol soluble fraction (ROBF) and ethanolic extract (ROEE) (80.26 ± 1.59% and 76.52 ± 1.26% inhibition, respectively) while moderate to minimum results were shown by ethyl acetate soluble (ROEF), *n*-hexane soluble (ROHF) and water soluble fractions (ROWF) (67.48 ± 0.49%, 58.08 ± 1.74% and 43.33 ± 0.62% inhibition, respectively). Such significant inhibition of tyrosinase may be due to the presence of bioactive constituents as revealed by LC-MS profile, such as hosloppin (Table 8), which showed maximum binding affinity as compared to standard kojic acid with tyrosinase enzyme and may be due to some other compounds in these extract/fractions.

Carbonic anhydrases are directly involved in electrolytes secretion, pH regulation, photosynthesis, tumorigenesis, biosynthetic processes, etc. [70]. Carbonic anhydrase inhibitors play an important role as anticonvulsant, antiglaucoma and anticancer agents. Recently, it has been proved that these inhibitors can be used for producing anti-infective drugs (antibacterial and antifungal agents) with novel mechanism of action [71]. For the first time, ethanolic extract (ROEE), *n*-hexane soluble fraction (ROHF), ethyl acetate soluble fraction (ROEF), *n*-butanol soluble fraction (ROBF) and water soluble fraction (ROWF) of roots of *R. odorata* were evaluated for their carbonic anhydrase inhibition activity. Ethyl acetate soluble fraction (ROEF) and ethanolic extract (ROEE) showed the highest % inhibition of enzyme than *n*-butanol soluble fraction (ROBF), *n*-hexane soluble fraction (ROHF) and water soluble fraction (ROWF) when compared to their respective standard, acetazolamide (standard). The % inhibition values observed for ethyl acetate soluble fraction and ethanolic extract were the highest (76.50 ± 0.67% and 72.59 ± 1.39%) which were comparable with acetazolamide (standard), i.e., 95.51 ± 1.29% while these values ranged from moderate to minimum for *n*-butanol soluble fraction, *n*-hexane soluble fraction and water soluble fraction (68.75 ± 1.69%, 56.64 ± 0.67% and 51.60 ± 1.13%), respectively. These results may be due to phytoconstituents identified by LC-MS profile, such as hosloppin (Table 9) which showed the maximum binding affinity among other compounds against carbonic anhydrase enzyme and may be due to some other compounds in the extract/fractions. This suggests the potential use of roots of *R. odorata* as a potential carbonic anhydrase inhibitor.

Toxicology tests can identify several of the problems that may result from the use of medicinal plants/herbs, particularly in vulnerable people [72]. Hemolysis is the rupturing of red blood cells (erythrocytes), which indicates the cytotoxic effects on red blood cells [73]. If the degree of hemolysis is greater than 30%, the plant extracts are deemed hazardous towards erythrocytes [74]. Table 4 presents the hemolytic activity of different extracts of roots of *R. odorata*. The ethanolic extract (ROEE) has the highest hemolytic percentage (18.80 ± 0.42%), followed by *n*-hexane soluble fraction (ROHF) (13.10 ± 0.77%), water soluble fraction (ROWF) (10.79 ± 0.51%), ethyl acetate soluble fraction (ROEF) (5.34 ±0.97%) and *n*-butanol soluble fraction (ROBF) has the lowest hemolytic activity (3.48 ± 0.69%). Overall, all five fractions have less than 30% hemolysis activity, so all fractions are nontoxic to humans and safe.

Molecular docking was carried out to evaluate ligand–enzyme interactions theoretically to understand the molecular basis of different biological activities of natural products. It provides better insights into the novel mechanism of action and binding affinity of active ligands against enzymes. To understand the inhibition potential of studied compounds and to compare enzyme inhibition results, 14 compounds from the LC-MS profile of ethanolic extract (ROEE) and *n*-butanol fraction (ROBF) (Azacridone–A, 4′,5,6,7,8-pentahydroxy-3′-methoxyflavone, 5-hydroxy-6-methoxycoumarin 7-glucoside, Piperolactam D, Artemidinol, Glyflavanone A, sloppin, 2-O-Caffeoylglucarate, Flazin, Isosinensetin, Euparin, N2-(2-carboxymethyl-2-hydroxysuccinoyl)arginine, Norvisnagin, 3′-Glucosyl-2′,4′,6′-trihydroxyacetophenone) along with thiourea (standard), kojic acid (standard) and acetazolamide (standard) were docked against urease, tyrosinase and carbonic anhydrase enzymes, respectively.

Conclusively, molecular docking results describe the interaction of urease, tyrosinase and carbonic anhydrase with the ligands Azacridone–A, 4′,5,6,7,8-pentahydroxy-3′-methoxyflavone, 5-hydroxy-6-methoxycoumarin 7-glucoside, Piperolactam D, Artemidinol, Glyflavanone A, Hosloppin, 2-O-Caffeoylglucarate, Flazin, Isosinensetin, Euparin, N2-(2-carboxymethyl-2-hydroxysuccinoyl)arginine, Norvisnagin and 3′-Glucosyl-2′,4′,6′-trihydroxyacetophenone characterized by LC-MS analysis, confirming our findings for the plant extract in terms of urease, tyrosinase and carbonic anhydrase inhibition assays.

## 4. Materials and Methods

### 4.1. Collection and Identification of Plants

Plants were purchased in the month of February 2019 from a local nursery located near Pattoki Bypass, Kasur, Punjab, Pakistan. They were authenticated as *Rondeletia odorata* by the Department of Botany, The Islamia University of Bahawalpur, Bahawalpur and were designated with reference no.167/Botany. One specimen was submitted in the Department of Botany herbarium, IUB, Bahawalpur for record.

### 4.2. Preparation of Plant Material

*R. odorata* plants were rinsed with tap water first followed by distilled water to remove the dirt on the surfaces of the plants. Plants were chopped into the aerial and root parts and then both plant parts were further cut into small pieces separately. They were shade-dried for about 720 h (30 days) and then pulverized into fine dry powders separately by using an electric grinder. Plant root powder was processed for further experimentation.

### 4.3. Preparation of Extract/Fractions

The shade-dried roots powder was extracted by maceration in 80% ethanol for a period of 15 days with occasional vigorous shaking. Filtration was completed using a Buchner funnel with mucilage cloth followed by Whatman filter paper no. 1. The filtrate was evaporated to dryness with a rotary evaporator by distillation at a temperature of 40 °C under reduced pressure [75]. The obtained extract was weighed and then suspended in 500 mL of distilled water. Aqueous extract was further treated successively with solvents of increasing polarity such as *n*-hexane, ethyl acetate and *n*-butanol using the soxhlet apparatus. All three fractions’ filtrates were evaporated to dryness by using a rotary evaporator at reduced pressure at 40 °C for *n*-hexane and ethyl acetate soluble fractions and at 55 °C for *n*-butanol soluble fraction. The dried fractions/extracts were weighed on an analytical balance (IRMECO), packed into the air-tight containers and kept at 4 °C until used for further experiments.

### 4.4. Phytochemical Analysis

#### 4.4.1. Preliminary Phytochemical Screening

*R. odorata* root extract and its various fractions were subjected to preliminary qualitative phytochemical screening tests to detect the presence of carbohydrates, proteins and amino acids, lipids, alkaloids, glycosides, flavonoids, tannins, phenols and saponins according to standard procedures described in [76,77].

#### 4.4.2. Determination of Bioactive Components

Total phenolic contents (TPC)

For determining TPC of roots extract/fractions, the Folin–Ciocalteu method [78] with minor modification was used. Stock solutions (1mg/mL) in methanol were made for all extract/fractions. Gallic acid (standard) with different concentrations (0, 50, 100, 150, 200 and 250 µg/mL) was also prepared in methanol. A gallic acid standard curve was drawn. Then, 200 µL each extract/fractions/standard and 200 µL Folin–Ciocalteu reagent were added to each Eppendorf tube. Each mixture was mixed by vortex mixture. After mixing, 800 µL of sodium carbonate was added and incubated for 2 h at room temperature. A total of 200 µL of mixture was placed into a 96 microreader plate. Absorbance was measured at 765 nm by BioTek Synergy HT (USA) microplate reader. TPC was expressed in milligrams of gallic acid equivalent/g of dry extract (mg GAE/g DE).

Total flavonoid contents (TFC)

TFC were measured by following the aluminum chloride method [78] with minor modifications. Stock solutions (1 mg/mL) in methanol were made for all extracts/fractions/standards. A mixture (1 mL extract/fractions + 4 mL de-ionized water + 0.3 mL NaNO_3_ + 0.3 mL of 10% AlCl3 solution) was subjected to mixing by vortex. Then, 2 mL 1M sodium hydroxide solution was poured into the above mixture. Incubation was completed at ambient temperature for 6 min. Finally, 2.4 mL de-ionized water was added and 200 µL of mixture was poured to the 96 microreader plate. Absorbance was measured at 510 nm using a BioTek Synergy HT (USA) microtiter plate reader. Quercetin was taken as the working standard. TFC was expressed in milligrams of quercetin equivalent/g of dry extract (mg QE/g DE).

### 4.5. In Vitro Pharmacological Evaluation

Antioxidant activities were examined using different methods and in vitro pharmacological studies were carried out for extract/fractions of roots of *R. odorata*.

#### 4.5.1. Antioxidant Screening

Antioxidant screening included two types of analyses: (1) radical scavenging analysis and (2) reducing power analysis. In both analyses, trolox was used as the working standard.

Radical scavenging assays

2,2′-azino-bis(3-ethylbenzothiazoline-6-sulfonic acid (ABTS) and 2,2-diphenyl-1-picrylhydrazyl (DPPH) assays were performed for the determination of radical scavenging potential of extract/fractions of roots of *R. odorata*. Procedures mentioned in [79] were used with slight modifications.

1.ABTS assay

First, 7.0 mM ABTS and 2.45 mM potassium persulfate were mixed and incubated at 25 °C in darkness for the formation of ABTS+ radical cation. Stock solutions of extract/fractions were adjusted so their absorbance showed 0.700 ± 0.02 at 734 nm. Then, 200 µL ABTS+ solution and 100 µL extract/fractions solutions were poured into a 96-well microtiter plate. Incubation of plate was completed at 25 °C for 30 min. Absorbance was measured at 734 nm using a BioTek Synergy HT (USA) microwell plate reader. Results were expressed as milligrams of trolox equivalents/g of dry extract (mg TE/g DE).

2.DPPH assay

First, 90 µL of DPPH solution was mixed with 10 µL of extract/fractions solutions individually in a 96 microtiter plate. Then, incubation of the 96 microtiter plate was maintained at 37 °C in darkness. Absorbance was measured at 517 nm using a BioTek Synergy HT (USA) microtiter plate reader. Results were written as milligrams of trolox equivalents/g of dry extract (mg TE/g DE).

Reducing power assays

Ferric-reducing antioxidant power (FRAP) and Cupric-ion reducing analysis (CUPRAC) assays were applied for determining reducing capacities of root extracts/fractions. These assays were performed by using the literature already available [79] with some minor modifications. Results were expressed in milligrams of trolox equivalents/g of dry extract (mg TE/g DE).

3.FRAP assay

A total of 50 µL of extract/fractions solution was added to 20 mM ferric chloride + 0.3 M reagent (1 mL) in acetate buffer (pH 3.6) + 10 mM C_18_H_12_N_6_ in 40 mM HCl. Incubation was completed at 25 °C for 30 min and absorbance was measured at 593 nm. Similarly, solution without the extract/fractions was regarded as blank and analyzed by the same procedure. 

4.CUPRAC assay

First, 0.1 mL of roots extract/fractions solutions were mixed with 200 µL C_2_H_7_NO_2_ buffer (1M, pH 7.0) + 200 µL C_14_H_12_N_2_ (7.5 mM) + 200 µL cupric chloride (10 mM). Then, this mixture was incubated for 30 min at ambient temperature. The absorbance of the mixture was measured at 450 nm. Similarly, the solution without the extract/fractions was regarded as blank and analyzed using the same procedure.

#### 4.5.2. Enzyme Inhibition Potential

Activities of ethanolic extract/fractions of *R. odorata* were evaluated for inhibiting the activity of urease, tyrosinase and carbonic anhydrase which is demonstrated as % inhibition. The detailed methodology is described below.

Urease inhibition assay

Urease enzyme inhibition assay was performed as described by [26] with minor modifications. Total volume of the assay mixture was 200 μL which contained 15 μL urease enzyme solution, 15 μL 1 M phosphate buffer solution (pH: 7) and 15 μL extract solutions (5 mg/mL each). All solutions were poured in sterilized 96-well ELISA microplates and incubated for 15 min at 37 °C. The 40 μL urea solution was then added as the reaction substrate and ELISA plate was re-incubated under the similar conditions. After incubation, the pre-read was measured at 630 nm. After taking the pre-read, 45 μL phenol solution and 70 μL alkali reagents were mixed in the reaction mixture. The ELISA plate was incubated again for 50 min at 37 °C. Absorbance was taken again at 630 nm and regarded as the post read. Thiourea was the working standard while the reaction system without roots extract/fractions was considered as the control. The % inhibitions of various test solutions were measured using the formula given below:

Inhibition (%) of urease enzyme = 100 − [(Abs. of Post Read − Abs. of Pre Read)/Abs. of Control] × 100

Tyrosinase inhibition assay

Tyrosinase inhibition potential was determined as stated previously in [27] with some modifications. Kojic acid was the standard and the reaction system without extract/fractions was considered as the control. Assay total volume was 200 µL which constituted 20 µL of enzyme solution, 10 µL of test solution made in DMSO (dimethyl sulfoxide) and 150 µL of phosphate buffer of pH 6.8. The ELISA plate with the reaction mixture was incubated for 10 min at 30 °C and then absorbance was measured at 480 nm and was regarded as the pre read. Then, the reaction was allowed to start by adding 20 µL of L-tyrosine as substrate and again the incubation of the micro plate with the reaction mixture was maintained at 30 °C for 30 min. Post read was noted by measuring the absorbance of reaction mixture at 480 nm and the experiments were completed in triplicates. The % inhibition of tyrosinase was assessed by applying the formula below:

Inhibition (%) of tyrosinase enzyme = 100 − [(Abs. of Post Read − Abs. of Pre Read/Abs. of Control] × 100

Carbonic anhydrase inhibition assay

Carbonic anhydrase inhibition was completed as stated in [28] with minute modifications. Acetazolamide was taken as the standard and the reaction system without extract/fractions was considered as the control. The assay total volume was 200 μL. A 140 μL volume of Tris-HEPES buffer of pH 7.4 with a 20 μL volume of carbonic anhydrase enzyme and a 20 μL volume of test solutions (concentration of 5 mg/mL each) were mixed in sterilized 96-well ELISA microplates and were incubated for 15 min at 25 °C. Absorbance was noted at 400 nm as the pre-read. Then, 20 μL of substrate which was 4-nitrophenol acetate was added, the microplate was re-incubated at the same temperature for 30 min, and the post read was determined on the same wavelength. All the experimentation was carried out in triplicates and % inhibition of CA was quantified by the formula given below:

Inhibition (%) of Carbonic anhydrase= 100 − [(Abs. of Post Read − Abs. of Pre Read)/Abs. of Control] × 100

#### 4.5.3. Hemolytic Activity

The hemolytic effect of roots extract/fractions was evaluated using [80] with slight modifications. First, 10 mL of blood from human volunteers was collected and then poured into a top-screwed EDTA tube and centrifuged for 5 min. The upper layer was separated out and red blood cells were washed many times with 10 mL cooled sterilized isotonic phosphate buffer saline (PBS) with a pH of 7.4. Washed cells were again suspended in 20 mL PBS and root extract/fractions with a concentration of 1 mg/mL each were added to this mixture separately and incubated at 37 °C for 60 min. The hemolysis rate was calculated by determining the absorbance of hemoglobin present in the supernatant at the wavelength of 540 nm. The 0.1% Triton X-100 was used as the positive control and PBS as the negative control. Hemolysis (%) was calculated by using the following formula. 

Hemolysis (%) = (Abs. of sample − Abs. of negative control)/Abs. of positive control × 100

### 4.6. UHPLC-ESI-QTOF-MS Analysis

The metabolic profile of ethanolic extract and *n*-butanol fraction of roots of *R. odorata* was analyzed by UHPLC-ESI-QTOF/MS analysis. It was performed on an Agilent-1290-infinity UHPLC system attached to an Agilent-6520-AccurateMass ESI-QTOF-MS. An Agilent Zorbax Eclipse Plus XDB-C18 column (2.1 × 150 mm in length, 3.5 μm in thikness) was used for separating the metabolites. The 0.1% formic acid in water was taken as mobile phase A while 0.1% formic acid in acetonitrile constituted the mobile phase B. A rheodyne type injector was used to inject 1.0 μL of injection volume with a flow rate of 0.5 mL/min and acquisition time of 25 min. The MS-scan was taken between 100 and 1000 employing electrospray ion sources in positive mode. Nitrogen gas was nebulizing and drying was completed at a flow rate of 25 and 600 L/h, respectively, with a drying gas temperature of 350 °C. Fragmentation voltage was adjusted to 125 V, whereas capillary voltage for analysis was 3500 V. The METLIN database was used for the identification of the phytoconstituents.

### 4.7. In Silico Molecular Docking Studies

It is a very beneficial tool in computer-aided drug design studies. First of all, different protein molecules (urease, tyrosinase and carbonic anhydrase) were taken from the Protein Data Bank (PDB) in PDB format with protein resolutions below 3 A°. The preparation of protein was completed in Discovery Studio 2021 Client. Different chains except A chain, water molecules and ligands already attached were removed from the protein molecules. Then, polar hydrogen molecules were added to proteins and saved as a Protein Data Bank file. Secondary metabolites chosen from liquid chromatography–mass spectrometry (LC-MS) analytical technique table and standard compounds were downloaded from the PubChem database in SDF (structure-data file) format. Then, prepared protein molecules were uploaded to PyRx software and were subjected to autodock and macromolecule options were made. The ligands were uploaded into PyRx from Open Babel for preparation of ligands. After that, the chemicals were converted to PDBQT format. Then, the grid box was formed in specific dimensions. Finally, interactions were visualized using the Discovery studio [81].

### 4.8. Statistical Analysis

Whole experimentation was completed in triplicates and results were represented as average ± S.D. (standard deviation). One way ANOVA was applied, followed by LSD test for comparing various study groups. Statistix version 8.1 was used for analyzing the results. *p* values < 0.05 were considered as significant values.

## 5. Conclusions

The present study revealed in vitro antioxidant % inhibition of urease, tyrosinase and carbonic anhydrase as well as hemolytic activity potential of *R. odorata* root extracts/fractions. In total, 28 and 20 compounds from ethanolic extract and *n*-butanol fraction, respectively, were identified through LC–MS analysis, which showed many pharmacological activities in in vitro experiments. A high binding affinity was observed for glyflavanone A and hosloppin in urease, tyrosinase and carbonic anhydrase inhibition. The computed binding energies of the compounds revealed that all the compounds had synergistic effects to prevent different diseases caused by these above-mentioned enzymes. Therefore, the findings of this study indicated that this plant is an excellent candidate for the treatment of ulcers, skin-related problems and diuresis. The species displayed promising results overall, but the tyrosinase inhibition activity was dominant. The medicinal and pharmacological potential of roots of *R. odorata* revealed that it is quite auspicious as a versatile therapeutic plant and should be further investigated.

## Figures and Tables

**Figure 1 molecules-27-04204-f001:**
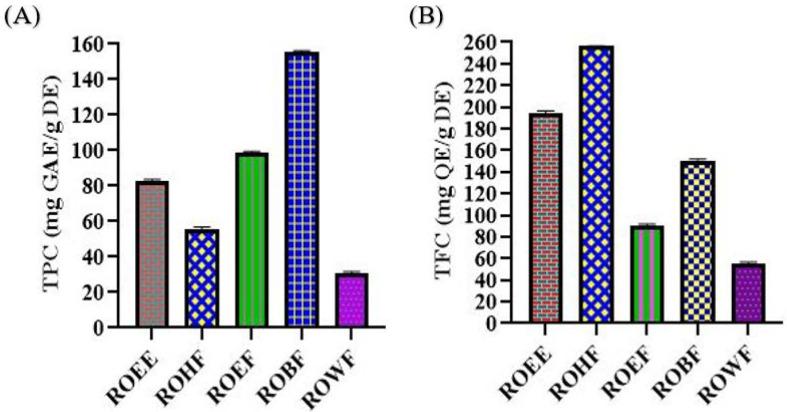
(**A**) Total phenolic contents (TPC) and (**B**) total flavonoid contents (TFC) of *R. odorata* root extract/fractions. ROEE: ethanolic extract; ROHF: *n*-hexane soluble fraction; ROEF: ethyl acetate soluble fraction; ROBF: *n*-butanol soluble fraction; ROWF: water soluble fraction; GAE: gallic acid equivalent; QE: quercetin equivalent; DE: dry extract.

**Figure 2 molecules-27-04204-f002:**
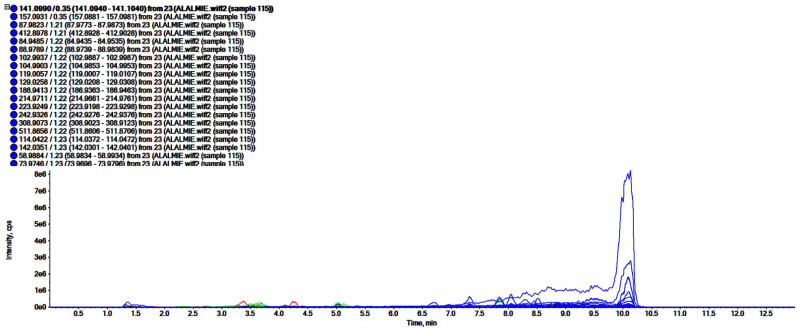
Total ion chromatogram of ethanolic extract of roots of *R. odorata* using UHPLC-ESI-QTOF-MS in positive electrospray ionization mode showing the chromatogram intensity against the acquisition time.

**Figure 3 molecules-27-04204-f003:**
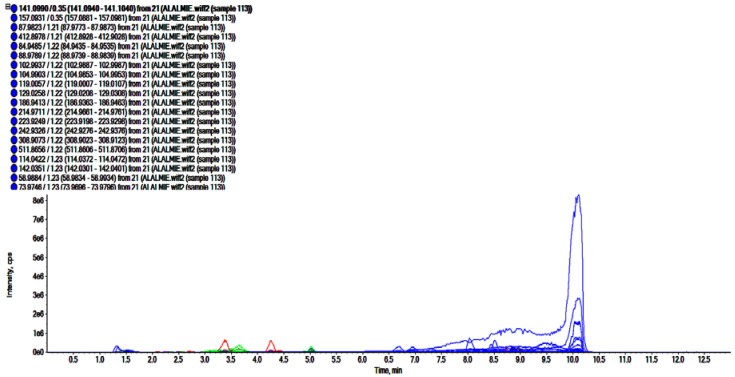
Total ion chromatogram of *n*-butanol fraction of roots of *R. odorata* using UHPLC-ESI-QTOF-MS in positive electrospray ionization mode showing the chromatogram intensity against the acquisition time.

**Figure 4 molecules-27-04204-f004:**
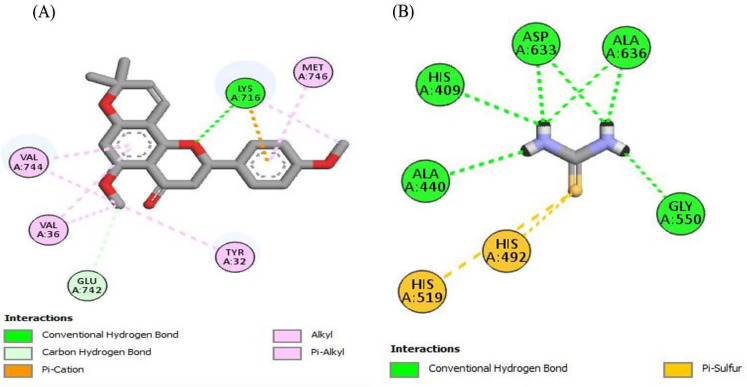
The 2D structured binding affinities of (**A**) Glyflavanone A and (**B**) Thiourea (standard) with urease enzyme.

**Figure 5 molecules-27-04204-f005:**
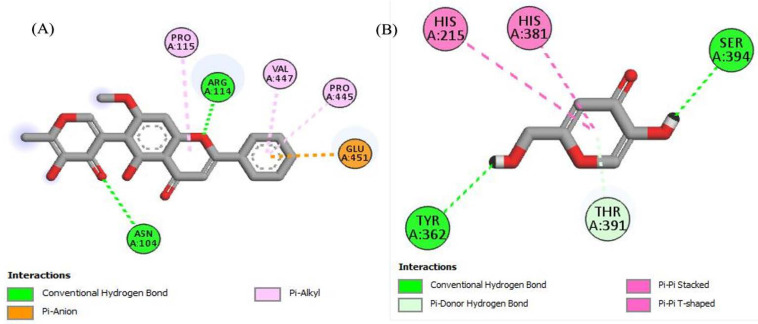
The 2D structured binding affinities of (**A**) Hosloppin and (**B**) Kojic acid with tyrosinase enzyme.

**Figure 6 molecules-27-04204-f006:**
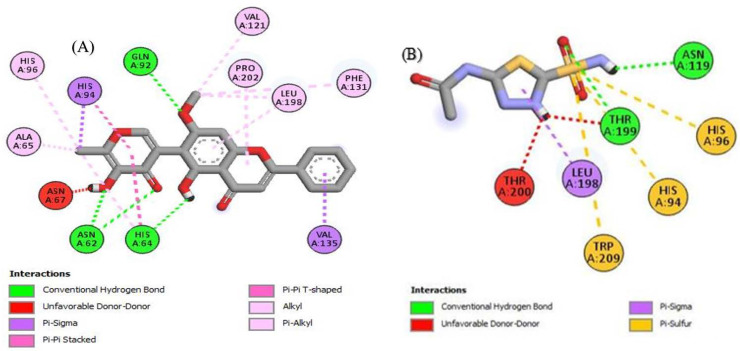
The 2D structured binding affinities of (**A**) Hosloppin and (**B**) Acetazolamide with carbonic anhydrase enzyme.

**Table 1 molecules-27-04204-t001:** Phytochemical screening of roots of *R. odorata* ethanolic extract and its various fractions.

	Metabolites	Tests	ROEE	ROHF	ROEF	ROBF	ROWF
Primary Metabolites							
1.	Carbohydrates	Molisch’s Test	−	−	−	+	−
		Fehling’s Test	−	−	−	+	−
		Iodine Test	−	−	−	+	−
2.	Proteins	Buerette Test	−	+	−	−	−
3.	Amino Acids	Ninhydrin Test	−	−	−	+	−
4.	Lipids	Saponification Test	+	−	+	+	−
Secondary Metabolites							
		Mayer’s test	+	+	+	+	+
1.	Alkaloids	Hager’s test	+	+	+	+	+
		Wagner’s test	+	+	+	+	+
2.	Glycosides	Erdmann’s Test	−	−	−	−	−
3.	Flavonoids	Alkaline Reagent Test	+	+	+	+	+
4.	Tannins	Lead Acetate Test	+	+	+	−	+
5.	Phenols	Ferric Chloride Test	+	+	+	+	−
6.	Saponins	Frothing Test	+	−	+	+	−

ROEE: ethanolic extract; ROHF: *n*-hexane soluble fraction; ROEF: ethyl acetate soluble fraction; ROBF: *n*-butanol soluble fraction; ROWF: water soluble fraction; +: present; −: absent.

**Table 2 molecules-27-04204-t002:** ABTS, DPPH, FRAP and CUPRAC values of extract/fractions of roots of *R. odorata*.

Extract/Fractions	ABTS (mg TE/g DE)	DPPH (mg TE/g DE)	FRAP (mg TE/g DE)	CUPRAC (mg TE/g DE)
ROEE	81.93 ± 1.45	197.85 ± 1.42	211.87 ± 1.60	255.02 ± 1.52
ROHF	49.25 ± 1.42	98.03 ± 1.45	172.73 ± 1.50	312.77 ± 1.03
ROEF	87.92 ± 1.44	111.03 ± 1.60	197.57 ± 1.31	201.15 ± 1.50
ROBF	82.66 ± 1.11	165.90 ± 1.73	239.92 ± 1.72	294.87 ± 1.84
ROWF	59.50 ± 0.80	51.47 ± 0.72	150.07 ± 1.59	145.26 ± 0.57

ROEE: ethanolic extract; ROHF: *n*-hexane soluble fraction; ROEF: ethyl acetate soluble fraction; ROBF: *n*-butanol soluble fraction; ROWF: water soluble fraction; TE: trolox equivalent; DE: dry extract.

**Table 3 molecules-27-04204-t003:** Urease, tyrosinase and carbonic anhydrase inhibition% of extract/fractions of roots of *R. odorata* (5 mg/mL) and standard drugs thiourea (0.375 mM), kojic acid (0.5 mM) and acetazolamide (0.1 mM), respectively.

Extract/Fractions	% Inhibition of Urease Enzyme	% Inhibition of Tyrosinase Enzyme	% Inhibition of Carbonic Anhydrase Enzyme
ROEE	73.39 ± 1.11	76.52 ± 1.26	72.59 ± 1.39
ROHF	53.97 ± 1.63	58.08 ± 1.74	56.64 ± 0.67
ROEF	66.36 ± 0.91	67.48 ± 0.49	76.50 ± 0.67
ROBF	70.29 ± 0.81	80.26 ± 1.59	68.75 ± 1.69
ROWF	45.69 ± 0.71	43.33 ± 0.62	51.60 ± 1.13
Standard	98.07 ± 0.74	98.59 ± 0.92	95.51 ± 1.29

All the values are represented as mean ± STD. ROEE: ethanolic extract; ROHF: *n*-hexane soluble fraction; ROEF: ethyl acetate soluble fraction; ROBF: *n*-butanol soluble fraction; ROWF: water soluble fraction.

**Table 4 molecules-27-04204-t004:** Hemolytic potential of roots extract/fractions of *R. odorata* (1 mg/mL) and standard Triton X-100 (0.1%).

Extract/Fractions	Hemolytic Potential (%)
ROEE	18.80 ± 0.42
ROHF	13.10 ± 0.77
ROEF	5.34 ± 0.97
ROBF	3.48 ± 0.69
ROWF	10.79 ± 0.51
Triton X-100 (standard)	93.07 ± 0.47

ROEE: ethanolic extract; ROHF: *n*-Hexane soluble fraction; ROEF: ethyl acetate soluble fraction; ROBF: *n*-butanol soluble fraction; ROWF: water soluble fraction.

**Table 5 molecules-27-04204-t005:** Tentative compound identification from ethanolic extract of roots of *R. odorata* by UHPLC-ESI-QTOF-MS analysis.

Sr. No.	Analyte Peak Mass	Retention Time (min)	Area/Height (%)	Tentative Identified Compounds	Chemical Class	Molecular Formula	Molecular Mass
1	217.0507	1.50	7.44	Norvisnagin	γ-Pyrone	C_12_H_8_O_4_	216.19
2	371.1216 M^+^Na^+^	1.51	7.62	N2-(2-Carboxymethyl-2-hydroxysuccinoyl)arginine	Arginine derivative	C_12_H_20_N_4_O_8_	348.31
3	367.1482	1.55	7.61	Glyflavanone A	Flavonoid	C_22_H_22_O_5_	366.4
4	389.1317 M^+^Na^+^	1.51	7.38	Ponganone III	Flavonoid	C_22_H_22_O_5_	366.4
5	304.0817 M^+^NH_4_^+^	2.43	10.83	3,7,8,4′-Tetrahydroxyflavone	Flavonoid	C_15_H_10_O_6_	286.24
6	309.1234	3.95	7.98	Azacridone-A	Pyridine derivative	C_18_H_16_N_2_O_3_	308.3
7	297.0745 M^+^Na^+^	4.08	10.88	Wyerone epoxide	Fatty acid	C_15_H_14_O_5_	274.27
8	329.0803 M^+^CH_3_OH^+^H^+^	3.95	11.48	Mono-trans-p-coumaroylmesotartaric acid	Ester derivative	C_14_H_14_O_8_	310.26
9	157.0345	4.01	8.93	3-(Acetylthio)-2-methylfuran	Ether derivative	C_7_H_8_O_2_S	156.20
10	225.0577	4.08	10.81	Hydroxyanthraquinone	Quinone	C_14_H_8_O_3_	224.21
11	217.0890	4.11	10.02	Artemidinol	Phenolics	C_13_H_12_O_3_	216.23
12	217.0796	4.18	6.76	Euparin	Aromatic	C_13_H_12_O_3_	216.23
13	230.0843	4.24	7.72	Pteleine	Alkaloid	C_13_H_11_NO_3_	229.23
14	229.0889 M^+^CH_3_OH^+^H^+^	4.33	10.85	Xanthone	Phenolics	C_13_H_8_O_2_	196.20
15	385.0680	4.40	8.18	2-O-Feruloylhydroxycitric acid	Phenolics	C_16_H_16_O_11_	384.29
16	296.0921	4.41	9.79	Piperolactam D	Alkaloids	C_17_H_13_NO_4_	295.29
17	373.0689	4.57	8.94	2-O-Caffeoylglucarate	Flavonoid	C_15_H_16_O_11_	372.28
18	375.1901	4.64	10.63	Spinochalcone C	Ketone	C_25_H_26_O_3_	374.5
19	387.1206	4.79	10.03	1-O-Sinapoyl-β-D-glucose	Flavonoid	C_17_H_22_O_10_	386.3
20	339.1007	4.72	7.73	Hydrojuglone glucoside; 1-Caffeoyl-4-deoxyquinic acid	Phenolics; Flavonoid	C_16_H_18_O_8_	338.31, 338.31
21	487.0970	4.94	10.20	Garciduol C	Aromatic	C_27_H_18_O_9_	486.4
22	393.0945	4.98	5.76	Hosloppin	Flavonoid	C_22_H_16_O_7_	392.4
23	568.2085	5.04	4.61	Neoacrimarine H	Ketone	C_33_H_29_NO_8_	567.6
24	542.2300 M^+^NH_4_^+^	5.15	9.10	Ligustroside	Phenolics	C_25_H_32_O_12_	524.5
25	331.1070	5.19	8.11	3′-Glucosyl-2′,4′,6′-trihydroxyacetophenone	Phenolics	C_14_H_18_O_9_	330.29
26	397.1406	5.48	8.58	Aloesol 7-glucoside	Phenolics	C_19_H_24_O_9_	396.4
27	667.2075	5.53	6.79	Tetramethylquercetin 3-rutinoside	Flavonoid	C_31_H_38_O_16_	666.6
28	405.1302	5.50	7.77	Calomelanol C	Phenolics	C_24_H_20_O_6_	404.12

**Table 6 molecules-27-04204-t006:** Tentative compounds identification of *n*-butanol fraction of roots of *R. odorata* by UHPLC-ESI-QTOF-MS analysis.

Sr. No.	Analyte Peak Mass	Retention Time	Area/Height	Identified Compounds	Chemical Class	Molecular Formula	Molecular Mass
1	333.0540	1.50	8.58	4′,5,6,7,8-Pentahydroxy-3′-methoxyflavone	Flavonoids	C_16_H_12_O_8_	332.26
2	216.0689	1.61	11.74	Robustine	Alkaloid	C_12_H_9_NO_3_	215.20
3	287.0563	2.40	13.81	7,8,3′,4′-Tetrahydroxyisoflavone	Flavonoids	C_15_H_10_O_6_	286.24
4	309.0871	2.74	5.34	Flazin	Alkaloid	C_17_H_12_N_2_O_4_	308.29
5	230.0817	3.49	7.66	Pteleine	Alkaloid	C_13_H_11_NO_3_	229.23
6	300.0872	3.51	5.91	Avenanthramide 1c	Phenolics	C_16_H_13_NO_5_	299.28
7	372.1059	3.90	12.76	Berberine chloride	Alkaloids	C_20_H_18_ClNO_4_	371.8
8	336.0857	4.01	11.59	Oxonantenine	Alkaloid	C_19_H_13_NO_5_	335.3
9	230.0843	4.23	7.26	gamma-Fagarine	Alkaloid	C_13_H_11_NO_3_	229.23
10	568.1942	4.61	7.25	Neoacrimarine H	Ketones	C_33_H_29_NO_8_	567.6
11	359.1472	4.62	7.94	Glicophenone	Diaryl	C_20_H_22_O_6_	358.4
					ethene derivative		
12	286.1446	4.69	6.61	Erysopine	Alkaloid	C_17_H_19_NO_3_	285.34
13	387.1206	4.80	11.90	8-Cinnamoyl-3,4-dihydro-5,7-dihydroxy-4-phenylcoumarin	Phenolics	C_24_H_18_O_5_	386.151
14	373.1261	4.81	8.47	Isosinensetin	Flavonoids	C_20_H_20_O_7_	372.4
15	329.1378	4.84	8.57	5-O-Methylleridol	Phenolics	C_19_H_20_O_5_	328.40
16	263.1291	4.86	8.21	Enokipodin D	Quinones	C_15_H_18_O_4_	262.30
17	301.0716	4.84	9.30	Scutevulin		C_16_H_12_O_6_	300.26
18	373.1047	4.90	9.72	Lophirone E	Phenolics	C_23_H_16_O_5_	372.4
19	371.0892	4.90	7.39	5-Hydroxy-6-methoxycoumarin 7 glucoside	Phenolics	C_16_H_18_O_10_	370.31
20	345.1376	5.03	6.66	Diosbulbin B	Diterpene lactones	C_19_H_20_O_6_	344.36

**Table 7 molecules-27-04204-t007:** Binding affinities and interactions of the examined compounds, isolated from roots of *R. odorata* against urease enzyme.

Ligand	Binding Affinity (Kilocalories/Mole)	Amino Acid Interactions					
Azacridone–A	−8	Van der Waals PHE:840; SER:834; ASN:580; SER:579; THR:830; THR:578; THR:829; PRO:576; ARG:646 GLU:642	Conventional Hydrogen Bond ARG:575		Pi-Sigma VAL:831	Pi-Pi Stacked PHE:838	Pi-Alkyl VAL:831
4′,5,6,7,8-pentahydroxy-3′-methoxyflavone	−8	Conventional hydrogen bond TYR:32; GLU:742; GLN:82; LYS:709	Carbon hydrogen bond VAL:81; VAL:744; ASP:730		Pi-Pi T-shaped TYR:32	Pi-Alkyl VAL:744; VAL:36	
5-hydroxy-6-methoxycoumarin 7-glucoside	−7.5	Conventional hydrogen bond HIS:593; ARG:609; GLY:550; ALA:440	Carbon hydrogen bond ASP:494			Pi-Pi Stacked HIS:593	
Piperolactam D	−7.9	Conventional hydrogen bond GLU:742	Pi-Cation LYS:716		Pi-Sigma THR:33	Pi-Alkyl VAL:36; ALA:37; PHE:712; VAL:744	
Artemidinol	−7	Conventional hydrogen bond LYS:716; TYR:32	Pi-Anion ASP:730		Pi-Pi Stacked PHE:712	Pi-Alkyl VAL:744; VAL:36; ALA:37	
Glyflavanone A	−9	Conventional hydrogen bond LYS:716	Carbon hydrogen bond GLU:742		Pi-Cation LYS:716	Pi-Alkyl VAL:744; VAL:36; TYR:32; MET:746	
Hosloppin	−8.2	Conventional hydrogen bond GLU:418	Carbon hydrogen bond ASP:730		Pi-Cation LYS:716; GLU:742	LEU:839; ALA:37; VAL:36; ALA:16	
2-O-Caffeoylglucarate	−7.3	Conventional hydrogen bond GLN:657; ARG:132; ASP:295; ARG:835; VAL:831; SER:834; ASN:836			Pi-Alkyl ALA:656; LYS:653; ALA:828		
Flazin	−8.7	Conventional hydrogen bond VAL:744; GLU:742	Pi-Anion ASP:730	Pi-Sigma VAL:36	Pi-Pi Stacked PHE:712; TYR:32	Pi-Alkyl LYS:716	
Isosinensetin	−7.5	Conventional hydrogen bond LYS:716	Carbon hydrogen bond GLU:742; VAL:744	Pi-Sigma THR:33	Pi-Alkyl VAL:36; TYR:32; ALA:16; PRO:743; LYS:745		
Euparin	−6.4	Pi-Sigma THR:33	Pi-Pi T-Shaped TYR:32		Pi-Alkyl VAL:744; VAL:36		
N2-(2-carboxymethyl-2-hydroxysuccinoyl)arginine	−6.5	Conventional Hydrogen Bond HIS:407; HIS:492; ALA:440; ALA:436; GLY:550; HIS:593; MET:588			Attractive charge ASP:633		
Norvisnagin	−6.6	Conventional Hydrogen Bond GLU:642	Carbon Hydrogen Bond THR:578		Pi-Sigma VAL:831	Amide-Pi Stacked SER:834	Pi-Alkyl PHE-838
3′-Glucosyl-2′,4′,6′-trihydroxyacetophenone	−6.4	Conventional Hydrogen Bond ARG:835; SER:834	Carbon Hydrogen Bond ASP:652		Pi-Anion ASP:295		
Thiourea (standard)	−3.4	Conventional Hydrogen Bond GLY:550; ASP:633; ALA:636; HIS:409			Pi-Sulfur HIS:519; HIS:492		

**Table 8 molecules-27-04204-t008:** Binding affinities and interactions of the examined compounds, isolated from roots of *R. odorata* against tyrosinase enzyme.

Ligand	Binding Affinity (Kilocalories/Mole)	Amino Acid Interactions			
Azacridone–A	−8.3	Conventional Hydrogen Bond VAL:129; SER:245; GLU:140	Pi-Pi T-Shaped PHE:244		Pi-Alkyl LEU:136
4′,5,6,7,8-pentahydroxy-3′-methoxyflavone	−8.9	Conventional Hydrogen Bond ARG:230; LYS:233; CYS:113; GLN:236	Pi-Sigma LYS:233		Pi-Alkyl PRO:115
5-hydroxy-6-methoxycoumarin 7-glucoside	−8.1	Conventional Hydrogen Bond ARG:230	Carbon Hydrogen Bond GLU:451; PRO:446	Pi-Pi T-Shaped TYR:226	Pi-Alkyl PRO:115; VAL:447
Piperolactam D	−8	Van der Waals SER:243; PHE:244; HIS:143; PHE:144; SER:245; ARG:130; ARG:131; VAL:129	Conventional Hydrogen Bond GLU:241	Pi-Anion GLU:140	Pi-Alkyl LEU:136; PRO:247
Artemidinol	−7	Conventional Hydrogen Bond CYS:113	Amide-Pi Stacked ARG:114	Alkyl ARG:118; PRO:242; VAL:126; PRO:115; LYS:233	
Glyflavanone A	−8.6	Conventional Hydrogen Bond TYR:226	Carbon Hydrogen Bond ASN:459	AlkylVAL:454; CYS:101; HIS:100; PRO:445	
Hosloppin	−10	Conventional Hydrogen Bond ASN:104; ARG:114	Pi-Anion GLU:451	Pi-Alkyl PRO:115; VAL:447; PRO:445	
2-O-Caffeoylglucarate	−7.1	Conventional Hydrogen Bond SER:106; TYR:226; ARG:230; GLU:232; GLN:236; ARG:114	Carbon Hydrogen Bond GLY:107; PRO:115	Pi-Alkyl LYS:233	
Flazin	−8.6	Conventional Hydrogen Bond GLU:451; ARG:114	Carbon Hydrogen Bond PRO:115; HIS:100	Pi-Anion GLU:451	Pi-Sigma PRO:445
Isosinensetin	−8.4	Carbon Hydrogen Bond HIS:143; SER:243	Pi-Anion GLU:241; GLU:140	Pi-Pi Stacked HIS:143	Alkyl LEU:136; PRO:247; PHE:144; PHE:244
Euparin	−6.8	Conventional Hydrogen Bond GLU:232		Pi-Alkyl PRO:115; LYS:233; LEU:229; TYR:226; ARG:230	
N2-(2-carboxymethyl-2-hydroxysuccinoyl)arginine	−7.3	Conventional Hydrogen Bond GLU:232; TYR:226; ARG:230; CYS:113; ARG:114; TRP:117			
Norvisnagin	−7.9	Conventional Hydrogen Bond LYS:233		Carbon Hydrogen Bond GLY:461	Pi-Alkyl ILE:128; PRO:115; LEU:229; ARG:230; TYR:226
3′-Glucosyl-2′,4′,6′-trihydroxyacetophenone	−8.2	Conventional Hydrogen Bond GLU:451; CYS:99; CYS:101; ARG:114; SER:106; PRO:445; PRO:446; THR:69		Carbon Hydrogen Bond HIS:100	Pi-Alkyl PRO:446
Kojic acid	−5.9	Conventional Hydrogen Bond SER:394; TYR:362		Pi-Donor Hydrogen Bond THR:391	Pi-Pi Stacked HIS:381; HIS:215

**Table 9 molecules-27-04204-t009:** Binding affinities and interactions of the examined compounds, isolated from roots of *R. odorata* against carbonic anhydrase enzyme.

Ligand	Binding Affinity (Kilocalories/Mole)	Amino Acid Interactions				
Azacridone–A	−6.7	Conventional Hydrogen Bond TYR:7	Pi-Anion GLU:239	Pi-Donor Hydrogen Bond ASN:11		Pi-Pi Stacked HIS:4
4′,5,6,7,8-pentahydroxy-3′-methoxyflavone	−7	Van der Waals THR:200	Conventional Hydrogen Bond ASN:119; GLN:92; ASN:62	Carbon Hydrogen Bond THR:199; SER:197	Pi-Pi T-Shaped HIS:94	Alkyl VAL:143; VAL:207; TRP:209; LEU:198
5-hydroxy-6-methoxycoumarin 7-glucoside	−7	Conventional Hydrogen Bond GLN:92; ASN:67; HIS:94		Carbon Hydrogen Bond HIS:64	Pi-Alkyl PRO:202	
Piperolactam D	−6.3	Carbon Hydrogen Bond TRP:5; GLY:6		Pi-Anion GLU:236; GLU:239	Pi-Alkyl PHE:231	
Artemidinol	−7	Conventional Hydrogen Bond GLN:92; THR:200		Pi-Sigma LEU:198	Alkyl VAL:143; HIS:96; VAL:121; TRP:209; HIS:94	
Glyflavanone A	−7.6	Conventional Hydrogen Bond THR:199	Carbon Hydrogen Bond ASN:119	Pi-Sigma LEU:198	Pi-Pi T-Shaped HIS:94; PHE:131	Pi-Alkyl VAL:121; TRP:209; HIS:96
Hosloppin	−7.8	Conventional Hydrogen Bond GLN:92; HIS:64; ASN:62	Donor-donor ASN:67	Pi-Sigma HIS:94; VAL:135	Alkyl Val:121; PRO:202; LEU:198; PHE:131; HIS:96; ALA:65	
2-O-Caffeoylglucarate	−7.3	Conventional Hydrogen Bond THR:199; GLN:92; ASN:67; ASN:62; THR:200	Carbon Hydrogen Bond HIS:94		Pi-Alkyl LEU:198; PRO:202	
Flazin	−7.3	Conventional Hydrogen Bond THR:199; THR:200	Pi-Sigma LEU:198	Pi-Pi T-shaped PHE:131; HIS:94	Pi-Alkyl VAL:121; PRO:202; VAL:135	
Isosinensetin	−6.3	Conventional Hydrogen Bond GLN:92	Pi-Sigma HIS:94			Alkyl HIS:96; HIS:64; LEU:198
Euparin	−6.1	Conventional Hydrogen Bond TYR:7; GLY:63; LYS:170		Carbon Hydrogen Bond HIS:4		Pi-Pi Stacked PHE:231
N2-(2-carboxymethyl-2-hydroxysuccinoyl)arginine	−6.6	Conventional Hydrogen Bond ASN:62; ASN:67; GLN:92; ASN:119; HIS:96		Carbon Hydrogen Bond HIS:64		Attractive ChargeGLU:106
Norvisnagin	−6.9	Conventional Hydrogen Bond THR:199; THR:200; GLN:92		Pi-Sigma LEU:198	Pi-Pi Stacked PHE:131; HIS:94	Alkyl TRP:209; VAL:121; VAL:143
3′-Glucosyl-2′,4′,6′-trihydroxyacetophenone	−6.4	Conventional Hydrogen Bond ASN:67; HIS:94; THR:199		Pi-Pi T-shaped HIS:64		
Acetazolamide	−6.2	Conventional Hydrogen Bond ASN:119; THR:199		Donor-donor THR:200	Pi-Sigma LEU:198	Pi-Sulfur HIS:96; HIS:94; TRP:209

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
