# Peer review of "Efficacy of Phytochemicals Derived from Roots of Rondeletia odorata as Antioxidant, Antiulcer, Diuretic, Skin Brightening and Hemolytic Agents—A Comprehensive Biochemical and In Silico Study"

_molecules, 2022, doi:10.3390/molecules27134204_

Round 1

Reviewer 1 Report

In the presented manuscript, a comprehensive phytochemical and biological evaluation of Rondeletia odorata was done.

The manuscript is well-organized and meets the high academic and methodological standards. The results are clear, consistent, and adequately presented. Moreover, it is thematically appropriate for the submitted Special issue. Although the references are mostly up-to-date, I suggest replacing some references in the Abstract section with the newer ones. Reference 25 should be checked, it seems incomplete. If you do not have complete article data, provide a link to the study and add the date of access.

It is not necessary to include both tables and figures containing the same data, so it is better to choose one presentation type, because the identical results are also repeated in the text of the Results section. The abbreviation explanation for TE and DE should be added to the legend of tables and figures, because some readers may not be familiar with these terms.

In my opinion, the manuscript provides useful information on the plant's chemistry and biopotential that may be served as a reference to the other species belonging to the genus Rondeletia.

Author Response

Reviewer 1

All the changes as indicated by the reviewer are highlighted green in the revised manuscript.

Comments and Suggestions for Authors

Comment: In the presented manuscript, a comprehensive phytochemical and biological evaluation of Rondeletia odorata was done. The manuscript is well-organized and meets the high academic and methodological standards. The results are clear, consistent, and adequately presented. Moreover, it is thematically appropriate for the submitted Special issue.

 Response: The authors are highly thankful for encouraging the work.

Comment: Although the references are mostly up-to-date, I suggest replacing some references in the Abstract section with the newer ones. 

Response: We appreciate the conscientious review. Some of the references have been replaced with recent references.

Comment: Reference 25 should be checked, it seems incomplete. If you do not have complete article data, provide a link to the study and add the date of access.

Response: Thank you for highlighting the point. We have rechecked the Reference 25. Link and date of the access has been added to the reference.

Comment: It is not necessary to include both tables and figures containing the same data, so it is better to choose one presentation type, because the identical results are also repeated in the text of the Results section.

Response: You are absolutely right. To avoid the duplication of data we have deleted table at one place and figures at another place. Figure 2 and 3 of the initial draft are now deleted. These two figures were presenting results of antioxidant assays and enzyme inhibition assay, which are also displayed in tabulated form (Table 2 & 3). Please kindly find the revised manuscript pages 4-7 for detail.

Comment: The abbreviation explanation for TE and DE should be added to the legend of tables and figures, because some readers may not be familiar with these terms.

Response: Thank you for the valuable comment. The abbreviation explanation for TE (trolox equivalent) and DE (dry extract) has been added in the corresponding sections of the manuscript. Please kindly find the changes in pages 5 & 6 of the manuscript for detail.

Comment: In my opinion, the manuscript provides useful information on the plant's chemistry and biopotential that may be served as a reference to the other species belonging to the genus Rondeletia.

Response: Thank you for the conscientious review.

Reviewer 2 Report

The proposed manuscript is generally well-organized. The experimental part included all necessary details and showed that the authors used proper methods for characterization of Roots of Rondeletia extracts and fractions. After LC-MS analysis authors identified Twenty-eight and twenty compounds for ethanolic extract and n-butanol soluble fraction respectively.

It should be emphasized that it takes effort and also implemented computational methods in this study. This makes the proposed manuscript more interdisciplinary, which is nowadays highly desired.

In summarizing I strongly recommend this manuscript for publication in molecules, but prior to publication authors should improve some details:

  1. Please highlight which activity of expects is the most promising; it can inspire others' research.

2. Please complete all gaps in table 5. You can find more general information about the chemical class, but it is still better to have free space.

3. Figure 4 is of poor quality.

4. In the case of figures 6-8, please provide the most important and move others to the supplementary material.

As a regular reader of molecules, I believe that the proposed work is above average.

Author Response

Reviewer 2

All the changes as indicated by the reviewer are highlighted green in the revised manuscript.

Comments and Suggestions for Authors

Comment: The proposed manuscript is generally well-organized. The experimental part included all necessary details and showed that the authors used proper methods for characterization of Roots of Rondeletia extracts and fractions. After LC-MS analysis authors identified Twenty-eight and twenty compounds for ethanolic extract and n-butanol soluble fraction respectively.

It should be emphasized that it takes effort and also implemented computational methods in this study. This makes the proposed manuscript more interdisciplinary, which is nowadays highly desired.

In summarizing I strongly recommend this manuscript for publication in molecules, but prior to publication authors should improve some details.

Response: Authors are highly thankful for the encouraging response and all the details mentioned have been addressed in the revised manuscript.

  1. Please highlight which activity of expects is the most promising; it can inspire others' research.

Response: Overall, the species showed promising results in all the in vitro activities but tyrosinase inhibition potential was found as the dominant. This has been mentioned and highlighted in the abstract and conclusion sections of the revised manuscript.

  1. Please complete all gaps in table 5. You can find more general information about the chemical class, but it is still better to have free space.

Response: Thank you for pointing out this. We are highly appreciating the thorough review of the manuscript. The gaps in Table 5 have been completed as per your suggestions.

  1. Figure 4 is of poor quality.

Response: Thank you for the valuable comment. Quality of Figure 4 (which is Figure 2 in the revised manuscript) has been improved.

  1. In the case of figures 6-8, please provide the most important and move others to the supplementary material.

Response: Thank you for your valuable suggestion. Following your instructions we have moved some figures to supplementary material.

Comment: As a regular reader of molecules, I believe that the proposed work is above average.

Response: Authors are highly thankful for your positive remarks.